Latent based temporal optimization approach for improving the performance of collaborative filtering

Al-Hadi Ismail Ahmed Al-Qasem 1
Sharef Nurfadhlina Mohd nurfadhlina@upm.edu.my 2
Sulaiman Md Nasir 2
Mustapha Norwati 2
Nilashi Mehrbakhsh 3
1 Faculty of Ocean Engineering Technology and Informatics, Universiti Malaysia Terengganu , Kuala Nerus , Terengganu , Malaysia
2 Faculty of Computer Science and Information Technology, Universiti Putra Malaysia , Serdang , Selangor , Malaysia
3 Faculty of Computing, Universiti Teknologi Malaysia , Skudai , Johor , Malaysia
Khan Faizal
Electronic publication date: 2020 Dec 21
Publication date: 2020
Volume: 6
Electronic Location ID: e331
Received 2020 Aug 17; Accepted 2020 Nov 16
Copyright: ©2020 Al-Hadi et al.
Copyright year: 2020
Copyright holder: Al-Hadi et al.
License: This is an open access article distributed under the terms of the Creative Commons Attribution License, which permits unrestricted use, distribution, reproduction and adaptation in any medium and for any purpose provided that it is properly attributed. For attribution, the original author(s), title, publication source (PeerJ Computer Science) and either DOI or URL of the article must be cited.
License URL: https://creativecommons.org/licenses/by/4.0/

Keywords: Temporal factorization, Recommender Systems, Collaborative Filtering, Drift, Decay, Matrix Factorization

Funding: The Asian Office of Airforce Research and Development (AOARD) through a project on Deep Recurrent Q Learning for Recommendation System This publication is funded by the Asian Office of Airforce Research and Development (AOARD) through a project on Deep Recurrent Q Learning for Recommendation System. The funders had no role in study design, data collection and analysis, decision to publish, or preparation of the manuscript.

==============================
Recommendation systems suggest peculiar products to customers based on their past ratings, preferences, and interests. These systems typically utilize collaborative filtering (CF) to analyze customers’ ratings for products within the rating matrix. CF suffers from the sparsity problem because a large number of rating grades are not accurately determined. Various prediction approaches have been used to solve this problem by learning its latent and temporal factors. A few other challenges such as latent feedback learning, customers’ drifting interests, overfitting, and the popularity decay of products over time have also been addressed. Existing works have typically deployed either short or long temporal representation for addressing the recommendation system issues. Although each effort improves on the accuracy of its respective benchmark, an integrative solution that could address all the problems without trading off its accuracy is needed. Thus, this paper presents a Latent-based Temporal Optimization (LTO) approach to improve the prediction accuracy of CF by learning the past attitudes of users and their interests over time. Experimental results show that the LTO approach efficiently improves the prediction accuracy of CF compared to the benchmark schemes.

Introduction

Recommendation systems are some of the most powerful methods for suggesting products to customers based on their interests and online purchases (Jonnalagedda et al., 2016; Lin, Li & Lian, 2020; Nilashi, bin Ibrahim & Ithnin, 2014; Nilashi et al., 2015; Zhang et al., 2020b). In terms of personalization of recommendations, one of the most prevalently used methods is collaborative filtering (CF) (Nilashi, bin Ibrahim & Ithnin, 2014; Sardianos, Ballas Papadatos & Varlamis, 2019; Nilashi et al., 2015; Wu et al., 2019). In CF, personalized prediction of products depends on the latent features of users in a rating matrix. However, the CF prediction accuracy decreases if the rating matrix is sparse (Zhang et al., 2020a; Li & Chi, 2018; Idrissi & Zellou, 2020). Several types of factorization techniques such as baseline, singular value decomposition (SVD), matrix factorization (MF), and neighbors-based baseline have been exploited to address the problem of data sparsity (Mirbakhsh & Ling, 2013; Al-Hadi et al., 2017b) by predicting the missing rating scores in the rating matrix. Similarly, various factorization-based techniques including the use of latent (Vo, Hong & Jung, 2020; Nguyen & Do, 2018) and baseline factors (Koenigstein, Dror & Koren, 2011) (such as SVD (Wang et al., 2019)) have been proposed to improve the recommendation accuracy. Nevertheless, an unaddressed problem is that a part of the rating scores is misplaced from its original cells while streaming into the memory. This misplacement decreases the meticulousness of the latent feedback.

A method based on ensemble divide and conquer (Al-Hadi et al., 2016) was adopted to solve the misplacement problem besides addressing the customers’ preferences drift and popularity decay. Integration of temporal preferences with factorization methods to solve the sparsity issue has yielded a better performance compared to basic factorization approaches (Al-Hadi et al., 2017b; Li, Xu & Cao, 2016; Nilashi et al., 2019; Nilashi, bin Ibrahim & Ithnin, 2014). The temporal dynamics approach (Koren, 2009) separates the time period of preferences into static digit of bins and extracts a universal weight according to the stochastic gradient descent method to reduce overfitting. Nonetheless, the learned universal weight using the temporal dynamics approach has limitations with respect to how it personalizes and represents the fluctuating temporal preferences.

The temporal interaction approach (Ye & Eskenazi, 2014) enhanced the effectiveness of CF recommender systems by combining the latent factors, short-term preferences, and long-term preferences. The shrunk neighbor approach is applied to obtain clients’ short-term feedbacks (Koren, 2008). This approach detects overfitting when there is a fluctuating scale in the rating scores. For example, in the rating matrix from the MovieLens dataset, the actual score is smaller than the predicted values. In this theory, the risk of each asset is measured with its beta value (which is the criterion of systematic risk). The fluctuating scale is in the range of 0–5, whereas the anticipated rating scores are [5.75; 6.11; 5.9; 7]. Compared to other temporal approaches (e.g., the short-term based latent technique (Yang et al., 2012)), the temporal interaction approach (Ye & Eskenazi, 2014) efficiently anticipates performance of CF. Nevertheless, problems such as drifting customers’ preferences and popularity decay (e.g., deterioration of marketability of goods) still pose a significant challenge (Ye & Eskenazi, 2014).

The long temporal-based factorization approach addresses the popularity decay issue (Al-Hadi et al., 2018b) while the short temporal-based factorization approach (Al-Hadi et al., 2017a) addresses the drift issue not solved by previous short-term based approaches. These temporal approaches improve the performance of CF but they are characterized by low accuracy. In view of the aforementioned, this paper presents a latent-based temporal optimization (LTO) approach to solve the significant limitations of these temporal approaches. As optimization algorithms have proven successful in various areas such as healthcare (Zainal et al., 2020)and document processing Al-Badarneh Amer (2016), we extend our earlier work (Al-Hadi et al., 2018a) and provide a detailed analysis of the proposed approach. The contributions of this paper are summarised below.

• A comprehensive review of CF-based recommender system techniques.

• A proposed LTO approach that minimizes overfitting and learns by integrating long and short-term features with the baseline and factorization features.

• An LTO approach that learns the drift in the users’ interests through an improved rating score prediction. This is achieved by integrating the long and short-term features of users and items with their baseline and factorization features.

• An LTO approach that solves the sparsity issue by combining the learning output of the overfitting, drift, and decay.

• A comparison of LTO’s performances with other factorization and temporal-based factorization approaches.

In summary, the proposed approach has superior performance as it improves the prediction accuracy in the CF technique by learning accurate latent effects of the temporal preferences of users. The novel features of the LTO approach are as follows:

• It provides a personalized temporal weighting which is incorporated in matrix factorization to reduce data sparsity error.

• It combines time convergence and personalized duration to accommodate consumer’s preferences drifting in the personalized recommendation system.

• It utilizes the bacterial foraging optimization algorithm (BFOA) to accurately learn the personalized temporal weights by regularizing the overfitted predicted scores in the rating matrix and to track the factors of drift and decay.

The rest of this paper is sectioned as follows: ‘Related works’ reviews the past works related to factorization approaches and temporal preferences. In ‘Latent-based temporal optimization approch’, LTO is elaborated, followed by experimental analysis in ‘Experimental Settings’. ‘Experimental results’ discusses the experimental results. The final section (‘Conclusion’) provides a summary and indicates possible future works.

Related Works

Collaborative Filtering

CF is a technique developed to make automated predictions (filtering) about the interests of a customer by gathering preferences or rating scores from several other customers (collaborating). The primary idea of the CF approach is that if a user (say X) shares an attitude with another user (Y) on a subject, X is more likely to share Y’s attitude on a different issue when compared to other randomly chosen users. CF is one of the most implemented techniques used in the design of recommendation systems due to its low computational requirement (Jonnalagedda et al., 2016; Sardianos, Ballas Papadatos & Varlamis, 2019; Alhijawi & Kilani, 2020).

It utilizes to find similar users or items and calculate predicted rating scores according to ratings of similar users. In addition, CF provides customized recommendations using the similarity values of customers and common preferences while the score of the active customer is placed in the rating score matrix. Changes are being made in the personalized recommendation to suggest products to customers based on their tastes. This constitutes a well-established methodology with a wide range of application.

In the CF technique, a forecast is achieved in three steps. The main step is estimating the values of similarity amongst common clients and the target customer with the use of similarity functions, such as the Cosine function (Nilashi et al., 2019; Alhijawi & Kilani, 2020). The rating scores supplied by the target client and the similarity values are applied in the next procedure to estimate the expected score of the product using the prediction function. The final step estimates the precision of the forecast by applying the root mean squared error (RMSE) function (Nilashi et al., 2019).

CF suffers from the data sparsity problem which occurs due to a soaring proportion of undetermined scores in the users’ voting matrix. This problem is solved using several prediction methods such as neighbors-based baseline (Bell & Koren, 2007) and matrix factorization (Koren, 2009; Nguyen & Do, 2018). However, these factorization-based methods do not address temporal issues such as the drift in users’ preferences and the popularity decay of products. This results in low prediction accuracy.

One of the most effective approaches for solving the data sparsity issue is MF (Koenigstein, Dror & Koren, 2011; Al-Hadi et al., 2017b). A few MF methods use mathematical formulae to combine hidden feedbacks of customers and products. The hidden feedback of customers, products, and baseline properties are incorporated in the formulae. Equation (1) forecasts the lost scores in the ranking matrix. (1) ℜ ˆui=μ+Bu+Bi+puqiT,

where ℜ ˆui is the predicted value for the sparse score, µ is the global rate of all rating scores, pu is the latent-feedback matrix of customers, qiT is the transpose latent-feedback matrix of products, and Bu and Bi are the observed deviations of customer u and product i, respectively.

To anticipate the sparse scores rating, µ, Bu, Bi, pu, and qiT are integrated in numerous mathematical equations such as those in temporal approaches (Koren, 2009; Ye & Eskenazi, 2014) and factorization methods (Al-Hadi et al., 2016; Han et al., 2018; Yuan, Zahir & Yang, 2019). For instance, the baseline factor and the distance between rating scores and baseline values of neighbors who supply their rating scores for each product are combined by the neighbors based baseline method (Bell & Koren, 2007) as presented in Eq. (2). (2) ℜ ˆui=Bu+∑x∈NisimxrxiBxi ∑x∈Nisimx,

where simx is the similarity rate of customer x with the target customer, N is the set of customers who rate product i, rxi is the rating score provided by user x for item i, and Bxi is the baseline value. Currently, the temporal recommendation methods are used to suggest products to customers at an appropriate time. These are applied in many prediction techniques to make an accurate forecast. Note that time is considered an important factor in making final decisions.

Temporal-based approaches

Time is a very important factor in learning customers’ interests and tracking products’ popularity decay. The temporal preferences with matrix factorization have been used to develop efficient collaborative-based schemes in addressing the issues of sparsity, drift, and decay. For example, the temporal dynamics approach (Koren, 2009) utilizes the factorization factors, bins (static temporal periods), and global weight to learn the temporal preferences and minimize the overfitted predicted scores. However, it neglects the fact that users’ preferences change over time. Hence, the overall weights are not accurate as a result of personalization.

The long-term preferences

Long-term preferences differ from short-term preferences with regards to how they are applied. In a session (i.e., a week, month, or season), the recorded preferences are considered short-term preferences. On the other hand, the baseline factors and the long-term preferences are exploited in the long-term approach (Ye & Eskenazi, 2014). This is expressed in Eq. (3), where τs, τe and τui are the first, last and current time that a product i is rated by a customer u, respectively. This approach addresses the drift in customers’ preferences over the long-term but it does not address the popularity decay of products. (3) ℜ ˆui=μ+Buτe−τuiτe−τs+Biτui−τsτe−τs+puqiT.

(4) ℜ ˆui=μ+Buωu+Biωi+puqiT2+GxiBuωu2+pu2+Biωi2+qiT2,

where Gxi is the weight of cluster x for item i that is updated by BFOA, pu and pu are the latent factor and norm of latent factor of customer u, while qiT and qiT are the latent factor and the norm of latent factor of product i, respectively. Moreover, the personal long-term factors are defined by ωu and ωi in Eqs. (5) and (6), respectively. (5) ωu=exp−τeu−τsuτeu,

(6) ωi=exp−τei−τsiτei,

where τeu and τsu are the last and the first time customer u provided a rating scores, and τsi and τei are the first and the last time the group of customers offers scores for product i, respectively. Nevertheless, the long temporal approaches (Al-Hadi et al., 2018b; Ye & Eskenazi, 2014) have not addressed issues such as the drift and the popularity decay by considering the short-term preferences. This lowers the prediction performance of the CF technique. The long temporal-based factorization approach (Al-Hadi et al., 2018b) learns the long-term preferences by integrating genres with factorization features to address sparsity and decay issues. However, this approach falls short of incorporating the drift in customers’ preferences which lowers the prediction accuracy of the CF.

The short-term preferences

The temporal dynamics approach (Koren, 2009) is used for predicting missing ratings by integrating the temporal weights with the different factorization factors. This approach minimizes the overfitted predicted scores during the optimization process using a global weight. However, it does not properly characterize personalized feedback. The short-term based latent method (Yang et al., 2012) learns the short-term preferences from the hidden feedback of neighbors’ preferences during a session. However, this approach is not a lasting solution, especially due to long-term, drift, and popularity decay. Similarly, the temporal integration approach (Ye & Eskenazi, 2014) integrates the long and short preferences with the baseline features to solve the drift issue. This approach is also limited by personalization, understanding the drift in users’ preferences, and items’ popularity decay over time.

The short-term based baseline (Ye & Eskenazi, 2014) incorporates the baseline values of neighbors during a session with other factorization factors as shown in Eq. (7). (7) ℜ ˆuit=Bui+∑j∈νu,truj−Bujϖijνu,t+puqiT,

where ϖij is the applied weight that decreases the overfitting predicted values, ν(u, t) is the set of products ranked by customer u during time interval t (e.g., the month of July), and ∑j∈νu,truj−Bujϖij shows the whole difference between the rating scores by customer u for a set of products during time t and the baseline values. Given the soaring ratio in the sparse values in the ranking matrix, the short-term methods are not efficient in learning short-term preferences.

Products and costumers’ preferences are learned through the short temporal-based factorization method (Al-Hadi et al., 2017a) to address the drift issue and improve the prediction accuracy of CF. However, product popularity decay is ignored in this approach. Short-term preferences are represented using the temporal convergence among the customers. These are exploited for the minimization of overfitting for the predicted rating scores as shown in Eq. (8), where ⊤ux is the temporal weight that is optimized according to the location of cluster number x that represents the short-term period. However, the prediction function of CF decreases due to the inability of the short-term methods to cover the drift and decay problems during the period. As such, the long and short-term preferences must be integrated to address all the issues in the recommendation system. (8) ℜ ˆui=μ+⊤uxBu+Bi+puqiT2+⊤uxBu2+pu2+Bi2+qiT2.

Summarily, the existing temporal-based approaches have addressed several limitations of recommender systems such as sparsity (Zhang et al., 2020a; Idrissi & Zellou, 2020; Chu et al., 2020), drift issue (Rabiu et al., 2020; Al-Hadi et al., 2017a) and time decay issue (Koren, 2009; Ye & Eskenazi, 2014; Al-Hadi et al., 2018b). Each reviewed approach in this article has one or two research gaps, e.g., learning the personalized features, the drift preferences, and the popularity decay. There is currently no approach that considers all these issues (Table 1). Therefore, this work introduces the LTO approach for learning the features related to all these issues.

Table 1 Comparison of temporal-based approaches according to the solved issues.

Temporal-based Approach	Short-Term	Long-Term	Sparsity	Drift	Decay	
Neighbors-based Baseline (Bell & Koren, 2007)			✓			
Temporal Dynamics (Koren, 2009)		✓	✓	✓	✓	
Ensemble Divide and Conquer (Al-Hadi et al., 2016)		✓				
Short-Term based Latent (Yang et al., 2012)	✓					
Temporal Integration (Ye & Eskenazi, 2014)	✓	✓		✓		
Long Temporal-based Factorization (Al-Hadi et al., 2018b)		✓	✓		✓	
Short Temporal-based Factorization (Al-Hadi et al., 2017a)	✓		✓	✓		

Latent-based Temporal Optimization Approach

The LTO approach addresses both long and short temporal preferences by using factorization to solve issues of preference drift and popularity decay (alg1). LTO applies RMSE, Cosine, and Prediction functions to assess the temporal preference representation. The key empirical setting of the temporal-based factorization method and the proposed solution framework are presented in Fig. 1.

BFOA is exploited to capture the preferences of a short duration. By applying k-means, the timestamp convergence deals with short durations in the time matrix. The number of clusters k is recognized based on the number of short durations in the entire period. Generally, bacteria cannot track the drift and the time decay perfectly during the short-term without considering the long-term. Therefore, the integration of the long and short durations represents the accurate solution for solving the limitations of the drift and the time decay.

Figure 1 Latent-based Temporal Optimization Framework.

In Fig. 2, we present an example of how to create the bacteria members by applying the k-means method. In this example, the clusters’ number is assigned 2 for each of the users’ and items’ features. Based on the time convergence between the products (columns) and customers (rows), four bacteria members are shown in Fig. 2.

Figure 2 An example of bacteria members initialization.

The standard BFOA is utilized in the LTO process to detect the temporal conducts of customers and products. BFOA members initialize the short-term weights using random values. The LTO approach changes these weights throughout the lifecycle of BFOA dynamically based on the positive effect it has on learning stages. The weights of bacteria members ⊤xu and ⊤yi are updated dynamically throughout the learning iteration. This provides a novel tracking of users’ interests in the items. The LTO uses Eq. (9) to reduce the overfitted predicted scores throughout the learning iteration. (9) Dui=μ+⊤xuBuωu+⊤yiBiωi+puqiT2,

where ⊤xu and ⊤yi indicate the short temporal weights indexed by clusters x and y. User u is indexed by cluster x and item i is indexed by cluster y. The values of ⊤xu and ⊤yi are updated in each iteration according to the positive effect in developing the accuracy prediction of the CF method. The vectors ωu and ωi are the long-temporal independent weights of customer u and product i while Bu, Bi, pu, qiT represent the baseline and factorization variables.

The second contribution of LTO is tracking users’ drifting interests. This is learned by focusing on the time associated with users’ interests as represented by Eq. (10). (10) Fu=⊤xuBuωu2+pu2,

where pu2 represents the norm value of user’s latent factor and ⊤xu is updated according to the positive effects of changing users’ interests throughout the learning process.

The third contribution of LTO is tracking the popularity decay of items throughout the learning process by focusing on the time popularity of items as shown in Eq. (11). (11) Gi=⊤yiBiωi2+qiT2,

where qiT2 is the norm factorization variable of items and ⊤yi is updated according to the improvement achieved through the learning iteration which affects the baseline values and norm factorization features of items. Furthermore, BFOA learns the significance of each short-term period by applying the RMSE (which acts as the fitness value).

These contributions are combined in Eq. (12) to predict the unknown values within the rating matrix. (12) ℜ ˆui=Dui+Fu+Gi.

The BFOA operates in three stages: chemotaxis, reproduction, and removal and distribution. The first stage involves seeking the closest nutrition source by the bacteria. This is accomplished by swimming or tumbling or alternating between swimming and tumbling to change direction during the generation. In this process, the flagella of the bacteria make clockwise rotations to choose another path so that rich nutrients in the surrounding can be obtained. The tumbling stage is expressed in Eq. (13). (13) τij+1,k,l=τij,k,l+Ci+θθitθi,

where τ is the short-term features of one bacterium, the variables i, j, k and l symbolize i-th bacterium at j-th chemotactic, k-th reproduction and l-th elimination and dispersal steps. Ci is the walk length in an irregular direction, Θi is a random value oncterium number i at j-th chemotactic, k-th reproduction and [-1,1], and ΘΘitΘi is the unit walk in the irregular direction.

Swimming follows tumbling whenever the flagella of bacteria make the counterclockwise rotation to move in a particular direction. The bacteria continue swimming in the same direction if the nutrients are rich and the alternation between tumbling and swimming is repeated until the chemotactic stage is complete. The swarming function utilizes a sensor to provide signals in a nutrient-rich environment. When the signal indicates a poor nutrient or a dangerous location, the bacteria shift from the center to the outward direction with a moving ring of the members. If the nutrient has a high level of succinate, the bacteria subsequently neglect aspartate attractant and concentrate in groups.

The bacteria provide an attraction signal for the all members so that they swim together. The bacteria move in a concentric pattern with a high density. The outward movement of the ring and the native releases of attractant constitute the spatial order (Kim & Abraham, 2007). The swarming stage is represented mathematically as shown in Eq. (14). (14) τij+1,k,l= ∑i=1S−dattr expwattrβ+ ∑i=1S−hrep expwrepβ,

where S denotes the number of bacteria and β is the summation of short-term features that can be learned by the members of bacteria i as shown in Eq. (15). The attractant depth dattr denotes the magnitude of excretion by a cell, attractant width wattr denotes how the chemical cohesion signal spreads, repellent height hrep and width wrep determine the size of optimization space where the cell is related to the dispersal of chemical signal. (15) β= ∑m=1Pτm−τmi2,

where P denotes the number of short-term features, τm denotes the short-term feature number m learned during the chemotaxis process while τmi is the short-term feature number m learned during the chemotaxis process by bacteria i.

In the reproduction stage, the health of bacteria is calculated according to the fitness value of each bacterium. The bacteria values are then sorted in an ascending order in the array. The fitness value provided by RMSE is extracted from the optimization area in the recommendation system (that is based on collaborative filtering). The lower half bacteria with poor foraging die and the upper half bacteria (having better foraging) are copied into two parts. Each part has the same values Al-Hadi et al. (2017a). This procedure keeps the bacterial population constant. The bacterial health can be calculated using Eq. (16). (16) Jhealthi= ∑j=1Nc+1Ji,j,k,l,

where Jhealthi is the healthy score of the short-term preference that can be learned by bacteria i, the number of chemotactic, reproduction and removal steps are j, k and l, respectively.

Provision for the possibility of the ever-changing status of a few bacteria is carried out in the third step (removal and distribution). Here, the rise order involves the arrangement and generation of the random vector. The bacteria are organized based on their health values. Moreover, the randomly generated locations are used to change the locations of the bacteria in the optimization domain. These locations are recognized as the prominent available locations. After the generation, the best result in each repetition is approved as the final (correct) result.

In this work, the BFOA is integrated with the k-means clustering algorithm and matrix factorization approach. The k-means acts as a clustering algorithm used to control the big optimization space based on members’ personal features. It is used to reduce the large number of members to a small number of clusters. These clusters are then controlled using a weight for each cluster to control the optimization domain. Applying the natural choice in the course of repeating generations, the BFOA decreases the number of poor foraging members and increases the number of rich foraging strategies (Al-Hadi, Hashim & Shamsuddin, 2011). After several generations, the poor foraging members are removed or transformed to skilled members.

Experimental Settings

The CF technique is used to predict the interest of an active user. This takes into account the calculated similarity values between the rating scores of common users (neighbours) and the active user. However, sparse rating scores in the rating matrix negatively affect the prediction accuracy of the CF technique. Thus, this research work aims to improve the prediction of sparse rating scores in the rating matrix of each active user. The data sparsity is an important issue which this research aims. The factorization approaches (including temporal-based factorization) are used to predict missing rating scores in the rating matrix which improves the prediction accuracy of CF. The percentage accuracy can be measured using RMSE function where the lower values of RMSE refers to the accurate predicted values for the missing rating score in the rating matrix and also refers to the heights accuracy prediction of the recommendation list of items that can provided to the active user.

Datasets

To demonstrate the performance of LTO, three real-world datasets are used: MovieLens, Netflix, and Epinions. Several experimental studies have utilized MovieLens [34], Netflix Prize [19], and Epinions [35] to predict the performances of recommendation systems. A brief description of the three datasets is given in Table 2. The customers of these datasets assign a rating score from 1 to 5 to the movie or product, where 1 to 2 indicate an unliked product and 3 to 5 indicate a liked product. In the concluding experiments, the sparsity level of each dataset is considered to show its effect on the prediction performance of these datasets. The sparsity level is computed by Eq. (17) (Abdelwahab et al., 2012), where #Rating is the score provided by users from 1 to 5 and #Total is the product of #Customers and #Products. (17) SparsityLevel=1−#Rating#Total.

Normalization

Data normalization is used in data transformation to reprocess the data with the aim of enhancing the precision and effectiveness of mining methods and distance calculations (Al-Hadi et al., 2016). In recommender systems, the scores of customers for products are within 0–5. However, this range may result in low prediction accuracy. Thus, the rating scores are normalized to a range (0–1) to reduce the prediction error. Table 3 shows the values of the original scores and the normalized scores.

Table 2 Experimental datasets.

	MovieLens	Netflix Prize	Epinions Trustlet	
#Customers	943	480,189	4,718	
#Products	1,682	17,770	36,165	
#Rating	100,000	100,480,507	346,035	
Sparsity level	0.93	0.98	0.99	
Date	1997–1998	1999–2005	1999	
#Periods	7 months	6 years	11 months	
Temporal vector	#seconds	#days	#months	
Products	Movie	Movie	Product	

k-means and BFOA setting

Table 4 shows the number of clusters for the k-means clustering method and the short-term periods for the three datasets. The MovieLens contains data for four periods (i.e., one day, one week, two weeks, and one month) while Netflix contains an additional two periods (i.e., one season and one year). For instance, the entire period of Netflix is about 2190 days which can be divided by 90 days to get 24 seasons. Here, the cluster number (i.e., 24) is assigned to represent the users’ activities throughout the temporal convergence of seasons. The k-means algorithm will divide the activities of users in the time matrix into 24 clusters. Similarly, the interest-time for items in the time matrix will be divided into 24 clusters. However, when the number of clusters is greater than the number of customers in some rating matrices (e.g., two weeks period), Netflix will not be appropriate for grouping by the k-means algorithm. The periods of one month and one season are applied by Epinions. The one-week period is not considered as it is not suitable for Epinions temporal feature (Al-Hadi et al., 2018b). The period of two weeks by Netflix is inappropriate for k-means clustering algorithm (Al-Hadi et al., 2017a). Therefore, in the Netflix dataset, three temporal periods are used (i.e., one year, one month, and one season). The BFOA factors and their values are determined according to the proper empirical execution of the LTO approach in Table 5. In addition, a value is selected from the numbers of clusters using the P parameter as demonstrated in Table 4.

Table 3 The normalization of the rating scores.

Type	Range	Rating Scores	
Original	[0–5]	0	1	2	3	4	5	
Normalized	[0–1]	0	0.2	0.4	0.6	0.8	1	

Table 4 The k-means clustering method and short-term periods for different datasets.

Datasets	# Days	Periods	# Clusters ( k)	Success clustering	
MovieLens	210	One month	7	✓	
		Two weeks	15	✓	
		One week	30	✓	
		One day	210		
Epinions	330	One month	11	✓	
		Two weeks	23	✓	
		One week	47		
Netflix	2190	One year	6	✓	
		One season	24	✓	
		One month	73	✓	
		Two weeks	156		

Table 5 The parameters values of BFOA.

Parameters	No	Parameters	No	
No. of bacteria groups S	6	Elimination-dispersal l	4	
The length of a swim	4	wrep	5	
Run length unit Ci	0.1	wattr	0.2	
No. of iteration	20	Probability of elimination-dispersal	0.25	
Optimum RMSE	0.01	dattr	0.1	
No. of chemotactic steps j	6	hrep	0.1	
Reproduction steps k	4			

RMSE value reduction

The LTO approach extracts the short-term features by deploying BFOA which uses the temporal convergence in the time matrix of a short duration. For instance, the number of months in each active user’s time matrix is considered to divide the time matrix into k-weighted clusters. The BFOA trains the clusters’ weights to reduce the overfitted predicted scores according to the smallest RMSE. The weights of a short duration are optimized based on the positive effects on the factorization factors. This way, the prediction accuracy of the CF technique is achieved.

The factorization factors are extracted from a rating matrix and fixed values are provided for all iterations of the optimization process. The swarming action of bacteria provides sensor values that are integrated with RMSE to guide the members of bacteria into the direction of the rich nutrient or to avoid the detrimental area. Short-term periods are determined from the time of all preferences in each experimental dataset and their effects are shown under two kinds of scoring scales. Table 6 is an example of drift learning based on overfitting values minimization. In this example, each row contains the active user’s ID and the RMSE values for 10 iterations. The lowest RMSE value is selected by ensemble selection and saved in the last column of this table. The optimum temporal weights are also saved according to the selected RMSE value. Average RMSE values for 10 iterations (column by column) are shown in the last row. The value in the last cell is the lowest average RMSE value of the test-set members (0.875). This will be used for comparing the prediction performance of LTO with the benchmark schemes.

The LTO learns the temporal weight of each user using the personality activation of the users who rated the set of items during the long-term. Similarly, the LTO learns the temporal weight of each item based on the personality activation of the set of users who rated the item during the long-term. The temporal weights of the long duration are incorporated in the baseline model to determine the interest of customers and the popularity of products. In addition, the short duration weights are learned by the LTO approach. This is achieved using minimized overfitting. LTO learns the drift and time decay features during the optimization process. This LTO approach improves the performance of the CF technique throughout the iteration loop by learning the accurate predicted sparse rating score values in the rating matrix which reducing the RMSE values. In the next section, the effect of LTO approach in learning the temporal features will be examined under the scoring of [0–5] and [0–1].

Experimental Results

This section discusses the performances of the benchmark and the proposed approaches for improving the CF prediction performance technique under two scoring scales which are [0–5] and [0–1]. The efficacy of LTO in resolving the decay and drift issues by reducing the RMSE values are also discussed. Table 7 shows the personal vectors of the Test-set matrices that can impact the experimental results. There are 31 rating matrices for MovieLens which are selected by the sequence 30, 60, 90, ..…, 930 for the Test-set. Each matrix in the Test-set has different numbers of rows and columns. The sparsity levels are compared with other matrices to provide unique results for each rating matrix. The average numbers of these matrices and their factors are used for performance evaluation in the experiments.

LTO approach under the scoring [0–5]

The LTO approach is applied to five short-term periods (which are a week, two weeks, a month, and a year) according to the tested dataset under the rating scale [0–5]. Figures 3–5 demonstrate the prediction accuracy while performing the iterations on the datasets. Figure 3 shows that the two weeks period in MovieLens has a higher prediction accuracy during 3–20 iterations compared to one week and one month. Users’ activities during the long and short duration preferences are best learned within the two weeks period. This makes it an accurate short-term preference.

Table 6 An Example of how LTO reduces RMSE values in Movielens under scoring [0-5].

User	Iteration number	Min RMSE	
	1	2	3	4	5	6	7	8	9	10		
1	0.827	0.814	0.801	0.797	0.793	0.786	0.782	0.782	0.780	0.777	0.777	
2	0.796	0.758	0.758	0.755	0.751	0.751	0.752	0.750	0.751	0.754	0.750	
3	1.274	1.255	1.245	1.231	1.231	1.229	1.243	1.226	1.229	1.222	1.222	
4	0.704	0.645	0.640	0.635	0.632	0.629	0.628	0.631	0.627	0.628	0.627	
5	1.128	1.116	1.058	1.036	1.018	1.018	1.009	1.005	1.002	1.000	1.000	
Avg. RMSE	0.946	0.918	0.900	0.891	0.885	0.882	0.883	0.879	0.878	0.876	0.875	

Table 7 Average personal vectors of the Test-set matrices.

Test-set	MovieLens	Netflix	Epinions	
Number of the rating matrices	31	20	5	
Avg. No. of customers	915	953	107	
Avg. No. of products	105	128	103	
Avg. No. of rating scores >0	17491	11662	517	
Avg. No. of total rating scores > = 0	97855	125183	10481	
Avg. percentage of SparsityLevel	79.18 %	89.12 %	94.59 %	

In Fig. 4, the period of one month in the Epinions dataset has a higher prediction accuracy during iterations 4 to 20 compared to the period of one season. Here, one month is an accurate short-term period. This period has the best short and long-term performances compared to one season. One year period in Netflix provides a greater prediction performance from iterations 1 to 5 in Fig. 5. Across iterations 5 to 20, the period of one year is more precise than one month but less than the period of one season.

The temporal attitudes are learned by the LTO approach over 20 iterations. This helps to achieve precise predictions by optimizing the temporal weights of each duration. The periods of Netflix are 6 years, 42 seasons or 73 months. Experimental results show that LTO, in one season, achieved the highest prediction performance compared to its predicted performance using the year and month periods. This is because the duration of a season is an intermediate between a month and a year. Moreover, various customers’ activities are performed therein.

Figure 3 LTO prediction accuracy improvement of CF for MovieLens.

Figure 4 LTO prediction accuracy improvement of CF for Epinions.

Figure 5 LTO prediction accuracy improvement of CF for Netflix.

LTO approach under the scoring [0–1]

This subsection demonstrates the normalizing effect on the performance of LTO for reducing the RMSE under the scoring [0-1]. Figures 6, 7 and 8 track the effects of the temporal vectors in improving the prediction accuracy of the CF using the LTO approach. Figure 6 indicates that the RMSE of MovieLens for a week is better than that of a month under the rating scale [0–1]. Additionally, the RMSE during the period of two weeks is the best compared to those of a week and a month. This emphasizes the significance of the two-week period in learning the drift of customers’ interests and products’ popularity time decay.

Figure 6 LTO prediction accuracy improvement of CF for MovieLens.

Figure 7 LTO improves accuracy prediction of CF for Epinions.

Figure 8 LTO prediction accuracy improvement of CF for Epinions.

Figure 7 shows the prediction accuracy using Epinions, Here, the period of one month has a significant effect on reducing the RMSE in iterations 3 to 20 compared to the effect of one season.

In Figure 8, the effect of a season using Netflix is equivalent to the effect of a year but more than the effect of a month during iterations 1 to 13. Iterations 13 to 20 has the sharpest accuracy prediction in the season compared to the period of one year and one month.

Figures 3–8 show the potential of BFOA in learning the temporal features by swarming in the dimensional time-space. The effects of the equivalent time periods show that the customers’ interests and the products’ popularity changed during these periods. The proposed approach will be evaluated by the current factorization and temporal approaches in the next subsection.

Comparison of the performances of CF, MF, and Temporal-based approaches

In this subsection, the LTO approach is evaluated by comparing its effectiveness in reducing RMSE values with other benchmark approaches. Both LTO and the benchmarks are used to predict the sparse scores as they all lower the RMSE values. Note that the lower the RMSE value, the higher the prediction accuracy of the CF approach. In Table 8, seven approaches are implemented to benchmark the prediction performance of LTO. The improvement in the prediction performance of the CF technique is represented by two scoring scales: [0–5] and [0–1]. First, the scores from 0 to 5 are provided by the users of the three experimental datasets, and the benchmarks are categorized into three parts. The first part contains the prediction accuracy of the CF technique by RMSE for the rating matrix of the active user without predicting the sparse rating scores. The second part contains the prediction accuracy of the CF by two factorization approaches which are Neighbour-based Baseline (Bell & Koren, 2007) and the Ensemble Divide and Conquer (Al-Hadi et al., 2016). These approaches are used to solve the sparsity issue and as well learn the accurate factorization features. From the evaluations, it is observed that the prediction performance of Ensemble Divide and Conquer is better than that of the CF and the Neighbours-based Baseline approach. However, the approaches in the second category have weaknesses in learning the overfitted predicted scores and temporal issues (such as drift and decay). For the third category, five temporal approaches are used to solve five issues i.e., sparsity, accurately learning latent features, overfitting, drift, and decay. The Temporal Dynamics (Koren, 2009) has a good prediction performance in solving these issues but it has a weakness in learning the personalized features using the equaled time slices. Temporal Integration using Netflix performs better compared to Temporal Dynamics and Short-Term based Latent approach. However, Temporal Integration has a weakness with respect to drift and decay. Short Temporal-based Factorization (Al-Hadi et al., 2017a) addresses all issues except popularity decay. It improves the prediction performance of the CF technique when compared to the above approaches using MovieLens and Netflix. The performance of the Short Temporal-based Factorization approach is lower than that of the Ensemble Divide and conquer approach using the Epinions dataset because the recorded timestamp factors are registered using the number of months only (which represent weak temporal features). Distinctively, the LTO approach addresses all issues including the limitations in the benchmark schemes. It improves the prediction performance of the CF technique through a perfect combination of various factorization and temporal features. It also tracks the drift of users and the decay of items throughout the learning process. Table 8 shows that LTO exhibits superior prediction performance when compared to all benchmarks.

Table 8 The RMSE of several prediction approaches using three datasets.

Approach	Scoring [0–5]	Scoring [0–1]	
	MovieLens	Epinions	Netflix	MovieLens	Epinions	Netflix	
CF	0.9573	1.0536	0.9983	0.1915	0.2107	0.1997	
Neighbors based Baseline (Bell & Koren, 2007)	0.9613	1.0562	0.9982	0.1923	0.2112	0.1996	
Ensemble Divide and Conquer (Al-Hadi et al., 2016)	0.9481	1.0351	0.973	0.1896	0.2069	0.1948	
Temporal Dynamics (Koren, 2009)	0.9514	1.0486	1.0173	0.1903	0.2097	0.2035	
Short-Term based Latent (Yang et al., 2012)	0.9613	1.0562	0.9982	0.1923	0.2110	0.1996	
Temporal Integration (Ye & Eskenazi, 2014)	0.9557	1.0563	0.9982	0.1912	0.2112	0.1996	
Short Termporal based Factorization (Al-Hadi et al., 2017a)	0.8716	1.0492	0.9704	0.1771	0.2088	0.1900	
LTO	0.7933	0.9887	0.8136	0.1642	0.199	0.1564	

The normalization process for rating scores has reduced the RMSE values by almost 80 % due to the percentage difference between the rating scores [0–5] and [0–1]. For example, the percentage difference of LTO approach in MovieLens is calculated using Eq. (18). (18) PercentageDifference=1−Scoringscale1−Scoringscale2Scoringscale1,

where Scoringscale1 is from 1 to 5 and Scoringscale2 is from 0 to 1. The percentage difference between 0.7933 and 0.1642 is 79.3%. Figure 9 indicates the high prediction accuracy achieved by the LTO approach for the three datasets when compared with the benchmark methods. Additionally, the graphs show the positive impact of the normalization in reducing the RMSE by around 80%.

Figure 9 The normalisation effects on the prediction accuracy of CF. LTO provides the highest accuracy prediction for the CF under scoring [0–1].

Comparison of the output prediction scores of CF and LTO

The CF technique utilizes the similarity function to calculate the similarity between the active user and the common users or neighbors. In the second stage, CF utilizes the prediction function according to the similarity values to recommend items to the active user. However, CF’s predicted scores are not accurate because of the sparsity values in the rating matrix. This is solved using the LTO approach. Table 9 is an example showing the improved accuracy of the predicted scores achieved by the LTO approach. In this example, the short duration is one year. Active users rate items from 1 to 5, where 1 and 2 indicate unlike items while 3, 4, and 5 indicate the liked items.

Table 9 Feedback prediction scores by CF and LTO approaches.

items	i1	i2	i3	i4	i5	i6	i7	i8	i9	i10	i11	i12	i13	i14	
Active user	1	4	1	3	1	2	2	1	5	2	5	2	5	2	
CF	2.9	2.9	2.9	3.1	3.3	2.7	2.8	2.5	2.9	2.4	2.9	2.4	2.9	2.9	
	R	R	R	R	R	R	R	R	R	N	R	N	R	R	
LTO	0.4	4.3	2.4	2.3	2.2	2.5	1.8	2.4	3.9	2.3	5	3	3.7	3.3	
	N	R	N	N	N	R	N	N	R	N	R	R	R	R	

The CF predicts rating scores from 2.4 to 3.3. This provides the active users with recommended and unrecommended items (denoted by R and N, respectively in Table 9). On the other hand, the LTO approach predicts rating scores from 0.4 to 5.0 which provides more accurate prediction compared to the CF. Figure 10 visualizes the output in Table 9 and indicates the high prediction performance of the LTO approach.

Figure 10 Feedback prediction scores by CF and LTO approaches.

Conclusion

The CF performance is affected by several factors including changes in the customer’s taste, time decay in the popularity of products, data sparsity, and the overfitting in the predicted rating scores. Prior research has attempted to enhance the CF’s prediction function by integrating the long and short-term preferences via the temporal interaction method (Ye & Eskenazi, 2014) with the factorization factors. However, the achievement is low. The goal of the long temporal-based factorization approach (Al-Hadi et al., 2018b) is to solve the popularity decay problem and understand the drifting taste of clients over the long-term. On the other hand, the main focus of the short temporal-based factorization approach is to understand the behaviors of customers and solve the drift issue in the short-term. Nonetheless, there are several limitations associated with predicting popularity decay issues as well as the drift in customers’ preferences over time.

To address these problems, the LTO approach presented in this paper integrates both short and long-term preferences. It utilizes the k-means and BFOA method which derived the fitness value by combining the signal and RMSE values. The swarming function represents the preferences of the short-term based on the sensitivity of bacteria to rich nutrients or dangerous signals. According to the empirical findings, a higher prediction precision is achieved by the LTO approach compared to the benchmark approaches. This is attributed to the temporal-based factorization approach and its ability to enhance the accuracy of the CF technique by understanding the temporal behaviors in both long and short preferences.

Possible extensions of this work include integrating the LTO approach with other factorization features such as neighbors’ latent feedbacks. This would contribute to addressing issues such as the cold start when recommending new items to active users. Besides, the genre’s features of movies can be integrated with the factors that are utilized by LTO approach for the purpose of addressing challenges of new items by the MovieLens and Yahoo! Music datasets.

Supplemental Information

Supplemental Information 1 LTO approach

Matlab Code files.

Click here for additional data file.

Additional Information and Declaration

Competing Interests

Author Contributions

Data Availability

The authors declare there are no competing interests.

Ismail Ahmed Al-Qasem Al-Hadi conceived and designed the experiments, performed the experiments, analyzed the data, performed the computation work, prepared figures and/or tables, authored or reviewed drafts of the paper, developed the solutions proposed, and approved the final draft.

Nurfadhlina Mohd Sharef conceived and designed the experiments, analyzed the data, prepared figures and/or tables, authored or reviewed drafts of the paper, supervised and lead the project, and approved the final draft.

Md Nasir Sulaiman and Norwati Mustapha conceived and designed the experiments, analyzed the data, authored or reviewed drafts of the paper, and approved the final draft.

Mehrbakhsh Nilashi analyzed the data, authored or reviewed drafts of the paper, and approved the final draft.

The following information was supplied regarding data availability:

Codes are available in the Supplemental Files.

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
