# Peer review of "Latent based temporal optimization approach for improving the performance of collaborative filtering"

_PeerJ Computer Science, doi:10.7717/peerj-cs.331_

## Round 0.1 · original submission · Minor Revisions

Revise the manuscript based on the reviewers comments.

·

Basic reporting

Paper is well introduced and written in clear English.

Experimental design

Experiments cover theoretical hypothesis.

Validity of the findings

Findings are valid as authors claims with given significance evidence.

·

Basic reporting

- The English language that used in this article is clear and unambiguous.
- The literature that are provided in the article are sufficient and up to date.
- clear methodology for tackling and solving the problem with good structure.
- The datasets and raw data that were used are real-world datasets, which are clear defined and cited.
- The presented result are met the hypotheses of the paper

Experimental design

no comment

Validity of the findings

no comment

Additional comments

- The term collaborative that is mentioned in the title is not being defined and used in the article. in deed is not illustrated to improve the performance of filtering
- The cluster formation is not explain well, how many clusters? based on what? etc.

Reviewer 3 ·

Basic reporting

no comment

Experimental design

no comment

Validity of the findings

no comment

Additional comments

The paper is well written, and it has the potential to change the way temporal drifting and sparsed datasets are treated. The authors have developed a method to apply a Latent-based Temporal Optimization (LTO) approach to improve the prediction accuracy of CF by learning the past attitudes of users and their interests over time. The methodology part is well explained. However, few issues are raised.
- the results need to be improved. The reasons behind each finding should be explained by the support of previous related studies or authors' justifications.
- there exist some typo errors in figures 9 and 10. Specifically, in fig 9, the label font type is not consistent with the ones used in other figures. Also, no scaling is provided at the horizontal axis. The authors may need to scale the axes appropriately. While in fig 10, the horizontal axis title is not very clear.

---

## Round 0.2 · accepted · Accept

All the reviewers comments were considered in the revised manuscript.
I am writing to inform you that your manuscript has been Accepted for publication. Congratulations!

·

Basic reporting

The work is clear, author contributions to stat-of-art is reasonable.

Experimental design

Experiments cover theory.

Validity of the findings

results are reasonable

·

Basic reporting

- The level of English language of the paper is good enough
- The paper is rich with the contents and has up to date literature references
- Paper is well structured.
- The significant result have met the objective

Experimental design

The experimental and methodology that presented in the paper are well defined

Validity of the findings

The finding of the paper and conclusions are well stated, which linked to objectives of research paper and supported the arguments

Additional comments

The contribution of the paper is a publishable values

Reviewer 3 ·

Basic reporting

No comment

Experimental design

No comment

Validity of the findings

No comment

Additional comments

The authors have done well to follow all comments and made all the necessary corrections.